# Involvement of Pyocyanin in Promoting LPS-Induced Apoptosis, Inflammation, and Oxidative Stress in Bovine Mammary Epithelium Cells

Hao Zhu [1,2], Wendi Cao [2], Yicai Huang [2], Niel A. Karrow [3] and Zhangping Yang [1,2,*]

1 College of Animal Science and Technology, Yangzhou University, Yangzhou 225009, China; mx120210848@stu.yzu.edu.cn
2 Joint International Research Laboratory of Agriculture and Agri-Product Safety, Ministry of Education of China, Yangzhou University, Yangzhou 225009, China; mz120221501@stu.yzu.edu.cn (W.C.); mz120221528@stu.yzu.edu.cn (Y.H.)
3 Department of Animal Biosciences, University of Guelph, Guelph, ON N1G 2W1, Canada; nkarrow@uoguelph.ca
* Correspondence: yzp@yzu.edu.cn

**Abstract:** Pyocyanin (PCN) is an extracellular toxin secreted by *Pseudomonas aeruginosa* (PA), which has redox capacity and disrupts the redox balance of host cells, affecting cell function and leading to cell death. The aim of this experiment was to compare the degree of apoptosis, inflammation, and oxidative stress of bovine mammary epithelium cells (bMECs) induced by lipopolysaccharide (LPS) and pyocyanin (PCN) and to examine whether PCN can promote the apoptosis, inflammation, and oxidative stress of bMECs induced by LPS. In this study, 1 μg/mL LPS and 1 μg/mL PCN were finally selected for subsequent experiments through dose-dependent experiments. In this study, cells were not given any treatment and were used as the control group (NC). The cells were treated with PCN or LPS individually for 6 h as the PCN group (PCN) or the LPS group (LPS), and the combination of LPS and PCN challenge for 6 h as the LPS + PCN (LPS + PCN) group. Compared with the control and LPS groups, PCN resulted in a significantly upregulated expression of genes related to pro-inflammatory (IL-6, TNF-α, MyD88), apoptotic (Bax, Caspase3, Caspase9), as well as protein expression of components in the TLR4/NF-κB signaling pathway (TLR4, p-p65, p65), and p53 signaling pathway (p-p53, p53, Caspase9) ($p < 0.05$). Moreover, the expression of genes and proteins was significantly upregulated after PCN treatment combined with LPS compared to either LPS or PCN challenge alone ($p < 0.05$). The stimulation of PCN combined with LPS significantly increased reactive oxygen species (ROS) and malondialdehyde (MDA) production in bovine mammary epithelial cells (bMECs), as well as decreased glutathione peroxidase (GSH-Px) and total antioxidant capacity (T-AOC). Moreover, cells in the LPS + PCN group aggravated oxidative stress and antioxidant inhibition in cells. In addition, the expression of the corresponding genes and proteins related to the Nrf2 pathway (Nrf2, HO-1) was significantly down-regulated in the PCN group as compared to the control group ($p < 0.05$). Altogether, PCN stimulation exacerbates inflammatory reactions, apoptosis, and oxidative stress reactions, as well as when combined with LPS challenge in bMECs. Therefore, this study indicates that PCN manifests a role in promoting apoptosis, inflammation, and oxidative stress and interacting with LPS to enhance more serious biological stress responses.

**Keywords:** pyocyanin; LPS; apoptosis; cellular inflammation; oxidative stress

## 1. Introduction

Mastitis is one of the four major diseases of dairy cows that leads to significant economic losses worldwide within the dairy industry [1,2]. The mechanisms of mastitis initiation are usually known as pathogenically infected via the invasion of pathogenic microorganisms [3,4]. The opening level of the teat canal allows pathogenic microorganisms

to enter the mammary tissue of dairy cows and provides conditions for the growth of pathogenic microorganisms, which is the primary cause of mastitis [5]. Studies display that there are more than 150 species of pathogenic bacteria responsible for the disease, among which environmental Gram-negative bacteria are often the cause of an acute inflammatory response that leads to severe clinical signs, such as *E. coli* and *P. aeruginosa* [6,7]. Moreover, bacterial infections result in the release of toxins that lead to persistent cellular damage in mammary cells. Furthermore, immune active cell migration into the mammary gland may disrupt the blood-milk barrier due to epithelial cell necrosis or apoptosis [8,9].

Lipopolysaccharide (LPS) is an endotoxin located in the outermost layer of Gram-negative bacteria's cell wall, which is produced when the bacteria reproduce or undergo lysis [10]. LPS is known to cause the infected host to produce a variety of inflammatory mediators, which may trigger a cascade reaction of inflammatory responses throughout the body [11]. Upon a Gram-negative bacterial invasion of an organism, LPS acts on host cells, instigating monocytes and macrophages to generate and secrete significant amounts of cytokines such as tumor necrosis factor (TNF)-$\alpha$, interleukin (IL)-6, and other proinflammatory factors [12]. This process can impair the homeostasis of cell metabolism, break the integrity of cell barriers, and cause systemic inflammatory responses, as reported [13–15]. Numerous studies have highlighted, in recent years, a linkage between LPS-triggered inflammatory signaling and the TLRs (Toll-like receptors) family [16]. It has been uncovered that the signaling pathways of TLRs, which can activate intracellular signaling cascades, are effective for downstream transcription factors presented in some of the effector genes [17]. The TLR signaling pathway correlates closely with a range of immune disorders and is critical in safeguarding mammalian hosts [18–20].

PCN is a redox toxin secreted by *Pseudomonas aeruginosa* [21]. It has been used as a stimulus to trigger inflammation and apoptosis in vitro [22]. PCN has been reported in several studies with respect to its toxic effects, which include the generation of intracellular oxygen free radicals leading to cellular oxidative stress [23]. PCN causes early lysosomal dysfunction, alters the permeability of mitochondrial membranes, activates Caspase3, and induces apoptosis in neutrophils [24].

PCN is a toxin secreted by *P. aeruginosa* while it is alive, while LPS is released only after *P. aeruginosa* has been lysed. The researcher has used mice and in vitro cell experiments to study the virulence of PCN in vivo and confirmed that PCN plays a key role in *P. aeruginosa* infection [25]. It was found that *P. aeruginosa* and its extracellular factor PCN can induce the production of Interleukin-8 (IL-8) in mononuclear macrophages through MAPK-NF-$\kappa$B and PKC-NF-$\kappa$B signaling pathways [26]. The study has shown that the LPS-induced inflammatory cytokine TNF-$\alpha$ can activate the NF-$\kappa$B pathway [27].

Research on the toxicity of PCN has mainly concentrated on human epithelial cells [28,29]. However, the effect of PCN on the cellular immune responses in bovine mammary epithelial cells (bMECs) has not been well characterized. The present study aimed to investigate the effect of PCN and LPS on the induction of apoptotic, inflammatory, and oxidative stress responses in bMECs, as well as the combined challenge of LPS and PCN on bMECs. The findings provide meaningful insights for uncovering the effect of *P. aeruginosa*-induced infection and improving the prevention of mastitis in dairy farms.

## 2. Materials and Methods

### 2.1. Chemicals

PCN with a purity greater than 98% was obtained from Aladdin (85665, Aladdin, Shanghai, China). The LPS employed in this study was commercially procured with a purity greater than 99% (L5293, Sigma, St. Louis, MO, USA).

### 2.2. Cell Culture Conditions

All the experiments were carried out over four to six generations of cultivated cells. bMECs were cultured in a medium containing 90% DMEM/F12 Basic (8119417 Gibco, Carlsbad, CA, USA) and 10% fetal bovine serum (FBS). In addition, 100 IU/mL penicillin

and 100 μg/mL streptomycin (Gibco, Carlsbad, CA, USA) were added to the medium to prevent contamination by environmental pathogens. Cells were cultured in the incubators (37 °C, 5% CO$_2$).

### 2.3. In Vitro Experimental Design

The concentration of LPS used in this study was 1 μg/mL [30]. Cells were cultured in complete medium (DMEM/F-12 heat-inactivated FBS) for 12 h. Formal treatments were as follows: complete medium as control (NC), complete medium pre-treatment followed by 1 μg/mL LPS, 1 μg/mL PCN, or 1 μg/mL PCN + 1 μg/mL LPS treatment of cells for 6 h (LPS, PCN, PCN + LPS).

### 2.4. Scanning of Bacterial Adhesion

To determine the effect of different concentrations of PCN on *E. coli* adhesion levels, *E. coli* (isolated from milk samples of mastitis) were co-cultured with a pHrodo green probe (Thermo Scientific, Waltham, MA, USA) prior to infection with bMECs. Next, $1 \times 10^6$ CFU of *E. coli* with green fluorescent labeling and increased doses of PCN (1 and 2 μg/mL) were co-cultured with bMECs at 37 °C for 3 h. Images were acquired using DMi8 Microsystems GmbH (Leica, Wetzlar, Germany).

### 2.5. Determination of Adherent Bacteria on Bovine Mammary Epithelial Cells

Cells were inoculated into 12-well plates and grown to 90% to start the experiment. Next, $1 \times 10^6$ CFU of *E. coli* (same strain used in step 2.4) and increasing doses of PCN (1 and 2 μg/mL) were co-cultured with bMECs for 3 h at 37 °C. The cells were washed with PBS. The co-culture suspension was diluted 10 times on an MH AGAR plate and incubated at 37 °C overnight to count the number of colonies.

### 2.6. Cell Viability

The experiment was carried out to determine the cell viability in the presence of different doses of PCN using the CCK-8 kit (Vazyme, Nanjing, China). Briefly, cells were inoculated in 96-well plates and incubated at 37 °C for 18–24 h. Then, cells were treated with different concentrations of PCN for 6 h. After discarding the culture medium, the culture medium containing 10% CCK8 was read and incubated at 37 °C for 1–4 h. OD values were recorded using an enzyme marker at 450 nm.

### 2.7. Detection of Apoptosis

The apoptotic effects of PCN and LPS on cells were detected using the Annexin V-FITC/PI apoptosis detection kit (Vazyme, Nanjing, China). Using the BD FACSCalibur$^{TM}$ flow cytometer (Franklin Lakes, NJ, USA) for analysis.

### 2.8. ROS and Oxidative Stress Index Detection

The level of reactive oxygen species (ROS) production in all groups of cells was determined using an intracellular staining kit containing labeled 2′,7′-dichlorofluorescein diacetate (DCFHDA) (Beyotime, Shanghai, China) and was captured by DMi8 Microsystems BmbH (Leica, Wetzlar, Germany). MDA was detected with the MDA test kit (Beyotime, Shanghai, China) and determined at 532 nm of absorbance. GSH-Px was detected with the GSH-Px test kit (Beyotime, Shanghai, China) and determined at 420 nm of absorbance. T-AOC was detected with the T-AOC test kit (Beyotime, Shanghai, China) and determined at 520 nm of absorbance.

### 2.9. RNA Isolation and Quantitative Real-Time PCR

Total RNA was extracted using an RNA isolation reagent (Vazyme, Nanjing, China). The quality of the exacted RNA was evaluated using a 2100 bioanalyzer (Agilent Technologies, Santa Clara, CA, USA). Using a reverse transcription reagent (Vazyme, Nanjing, China) to transcribe RNA into cDNA. qRT-PCR was performed using real-time fluorescent

quantitative reagents (Vazyme, Nanjing, China) and the QuantStudio 3 Real-Time PCR System (Thermo Fisher Scientific, Waltham, MA, USA). Primers for caspase9, caspase3, Bax, Bcl-2, TLR4, MyD88, TNF-α, IL-6, Nrf2, Keap1, HO-1, Gpx1, UXT, RPS9, and GAPDH were designed using Primer 5.0. Primer sequences are listed in Supplementary Table S1. The $2^{-\Delta\Delta ct}$ method was used to calculate the relative mRNA expression of the target genes.

### 2.10. Western Blotting

Total proteins from bMECs were extracted using RIPA buffer (Biotek, Shanghai, China). Equal amounts of proteins were separated on polyacrylamide gels (ACE, Nanjing, China). The isolated proteins were transferred to the pvdf membrane (Millipore, Billerica, MA, USA). Isolated proteins were incubated with primary antibodies (Cell Signaling Technology, Danver, MA, USA) overnight at 4 °C and tailored according to the molecular weight of the target protein. Bax, Bcl-2, phospho-p53, p53, Caspase9, TLR4, MyD88, phospho-p65, p65, IL-6, Keap1, Nrf2, HO-1, and GAPDH antibodies were purchased from Cell Signaling Technologies (#41162, #4223, #82530, #2527, #9504, #14358, #4283, #3033, #8242, #8047, #20733, #86806, #2118). The primary antibody was diluted 1:1000. All antibodies in previous studies were shown to bind specifically to bovine protein [31]. The blot was washed moderately with TBST and incubated with the secondary antibody (#7074, CST); the secondary antibody was diluted 1:5000. Results were analyzed and quantified using Image J software 1.53e (LOCI).

### 2.11. Immunofluorescence

bMECs were inoculated into 12-well plates using crawler slides. After treatment selection, cells were fixed with 4% paraformaldehyde for 15 min, rinsed with PBS, and incubated with cell permeate for 15 min at room temperature. The cells were then incubated with bovine serum albumin (BSA) (5%) for 1 h at 37 °C and incubated with primary antibodies overnight at 4 °C. Cells were then stained with an indole tricarbocyanine (Cy3)-labeled secondary antibody. After three washes with PBS, the cells were stained with DAPI (1 mg/mL, D8417, Sigma-Aldrich, St. Louis, MO, USA) for 6 min at room temperature, and the images were captured using a DMi8 Microsystems GmbH (Leica, Wetzlar, Germany).

### 2.12. Statistical Analysis

SPSS 27.0 (IBM Inc., New York, NY, USA) was used for data analysis. Data were expressed as mean ± standard error (mean ± SEM) and analyzed using one-way ANOVA and Dunnett's post-hoc test. $p$-values < 0.05 were considered statistically significant differences. The experiment was repeated three times.

## 3. Results

### 3.1. PCN Facilitated the Adhesion of Escherichia coli to Host Cells

The number of *E. coli* adhering to the surface of bMECs increased when PCN was added, as a result of increased exposure of green fluorescent *E. coli* to both 1 and 2 μg/mL of PCN (Figure 1A). Furthermore, the growth determination of adherent bacteria on MH agar was measured as shown in Figure 1B. After treatment with 1 and 2 μg/mL of PCN, the number of *E. coli* adherent to bMECs significantly increased compared with the *E. coli* group ($p < 0.05$).

### 3.2. Effect of PCN and LPS on Cell Viability

The bMECs were treated with 0, 1, 2, 3, 4, and 5 μg/mL of PCN combined with 1 μg/mL of LPS for 6 h, respectively. The effects of different doses of PCN combined with 1 μg/mL of LPS on cell viability were determined using CCK-8. The viability of bMECs was significantly decreased by the stimulation of LPS at 1 μg/mL combined with different doses of PCN as compared to the NC group ($p < 0.05$). The viability of bMECs shows a decreasing trend with increasing PCN concentration, whereas no significant difference was found in cells treated with a combination of 1 μg/mL PCN and LPS as compared to the

LPS group. Therefore, the combination of 1 μg/mL of LPS and 1 μg/mL PCN was selected for the subsequent experiments (Figure 2).

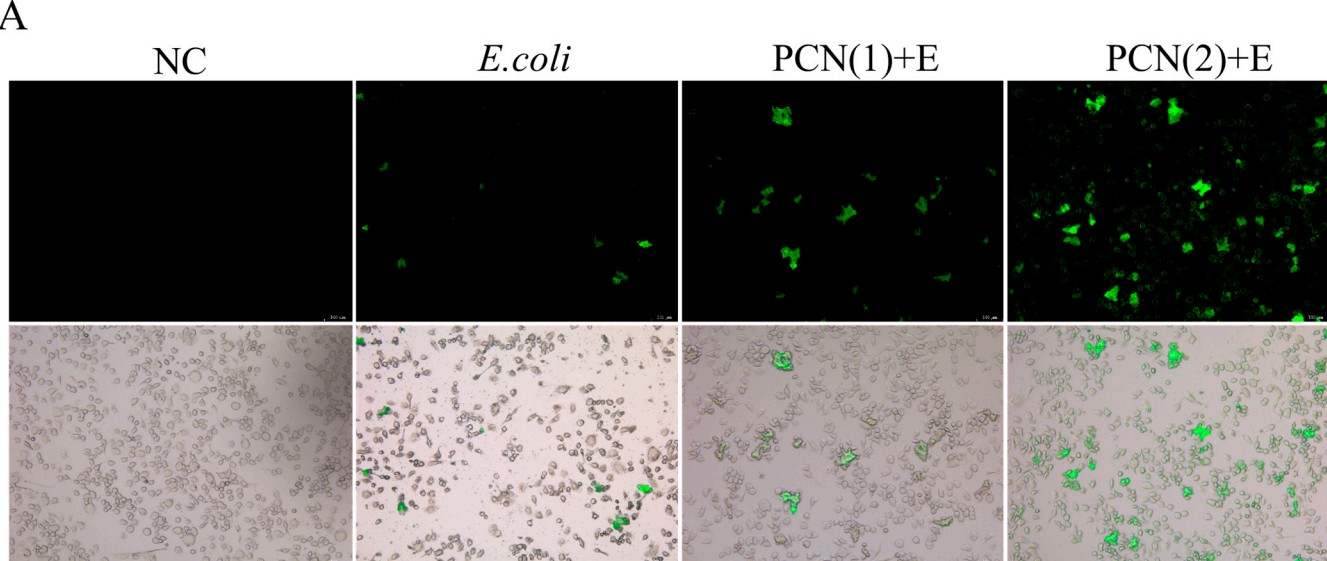

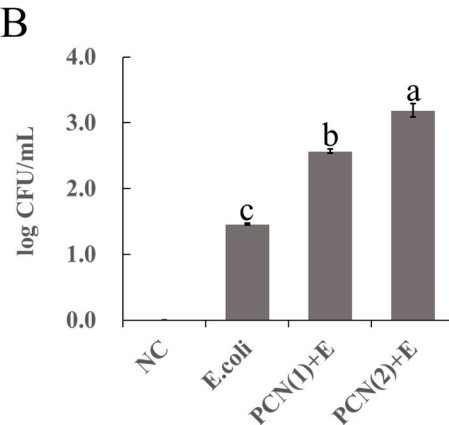

**Figure 1.** The promotion of the adherence property of *E. coli* in bMECs by PCN supplementation. (**A**) Image of bacterial adhesion in bMECs. Cells were infected with bacteria for 3 h, along with supplementation of PCN at doses of 1 and 2 μg/mL. *E. coli* were labeled with pHrodo (green). Scale bar = 1.00 mm. (**B**) Counting for adhered bacteria in bMECs. Cells were seeded on the 12-well plates, and $1 \times 10^6$ colony-forming unit (CFU) bacteria and increasing doses of PCN at 1 and 2 μg/mL were cocultured with bMECs at 37 °C for 3 h. The final colony number was measured after incubation at 37 °C for 18 h. Data are presented as mean ± SEM. The results are representative of three independent experiments. The letters in superscript indicate that the difference between groups was significant ($p < 0.05$). Groups with different letters indicate significant differences between the groups ($p < 0.05$), and groups with the same letter indicate no significant differences between the groups ($p > 0.05$). Data are representative of three independent replicates. bMECs, bovine mammary epithelial cells; PCN, pyocyanin; E, *E. coli*.

### 3.3. The Induction of Both PCN and LPS-Accelerated Apoptosis in bMECs

To investigate the effects of PCN, LPS, and combinations of LPS and PCN on the apoptotic level of bMECs and to understand the pathway that was affected in responses to stimulations, we measured apoptosis by flow cytometry under different treatments and detected apoptosis-related mRNA and protein expression. Compared with the control group, apoptotic levels in cells challenged with LPS were significantly promoted ($p < 0.05$). Moreover, the number of apoptotic cells was significantly increased by the addition of PCN as compared with the LPS group, indicating that PCN promotes apoptosis more

effectively than LPS, and the co-induction of cells by PCN and LPS significantly increased apoptosis as well (Figure 3A). Compared with the control group, the mRNA expression of the Cyt-C, Bax/Bcl-2, Caspase3, and Caspase9 genes in LPS or PCN-induced cells was upregulated ($p < 0.05$). Furthermore, the mRNA expression of Bax/Bcl-2, Cyt C, caspase3, and caspase9 was significantly increased in the PCN + LPS group compared to either the LPS or PCN groups (Figure 3B). In addition, the expression levels of p53-type pathway-related proteins in the PCN + LPS groups were significantly higher than those in the LPS or PCN groups (Figure 3C). The results consistently show a significant increase in p53 expression in cells treated with a combination of both compounds as compared to LPS or PCN alone (Figure 3D).

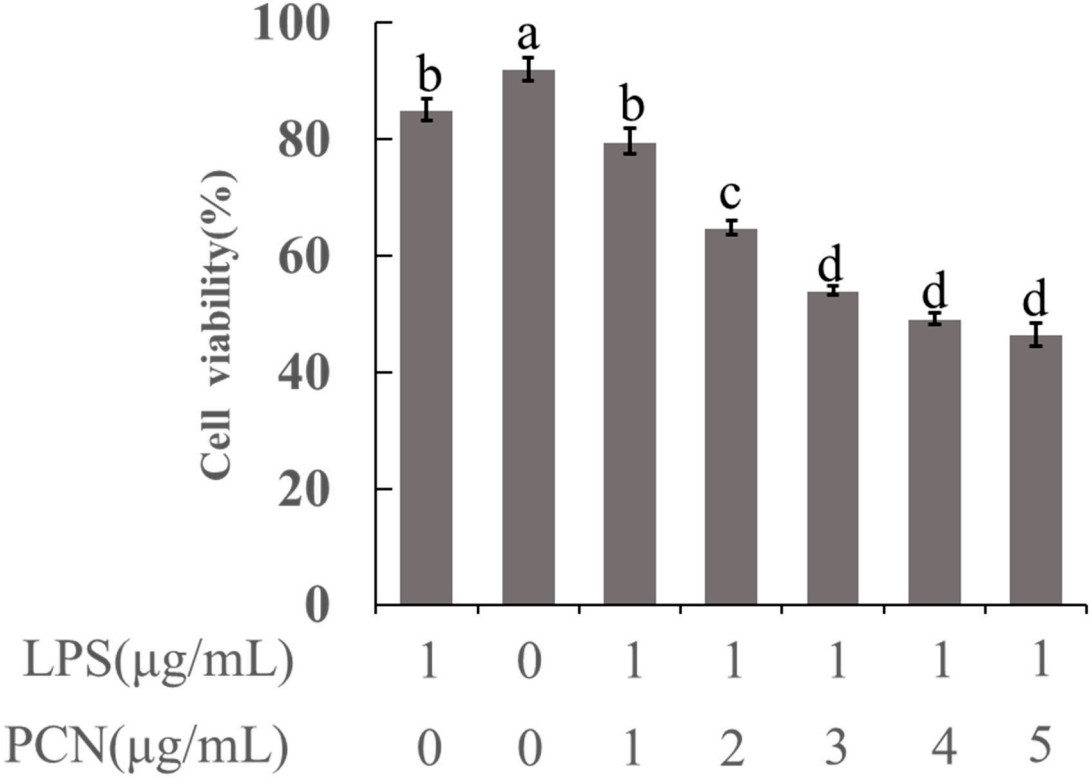

**Figure 2.** Cell viability treated with different concentrations of PCN. Cells were treated with PCN at concentrations of 0, 1, 2, 3, 4, and 5 μg/mL for 6 h and with LPS at concentrations of 1 μg/mL for 6 h. Data are presented as mean ± SEM. The results are representative of three independent experiments. The letters in superscript indicate that the difference between groups was significant ($p < 0.05$). Groups with different letters indicate significant differences between the groups ($p < 0.05$), and groups with the same letter indicate no significant differences between the groups ($p > 0.05$). Data are representative of three independent replicates. LPS, lipopolysaccharide; PCN, pyocyanin.

*3.4. Stimulation of PCN Combined with LPS Exacerbated the Inflammatory Response in bMECs*

We examined the expression of mRNAs and proteins associated with pro-inflammation. The mRNA expression of TLR4, MYD88, IL-6, and TNF-α was up-regulated in both LPS- and PCN-induced cells as compared to the control group ($p < 0.05$). Moreover, the PCN + LPS group significantly increased the expression of these genes compared to either the LPS or PCN group ($p < 0.05$) (Figure 4A). For protein expression, the components related to TLR4 signaling (TLR4, MyD88) were significantly up-regulated by challenging the cells with a combination of PCN and LPS as compared to PCN or LPS alone ($p < 0.05$). Plus, NF-κB signaling was also changed in cells by the treatment of PCN, LPS, or a combination of PCN and LPS. As a result, the ratio of phosphorylated p65 to total p65 was significantly increased in the PCN + LPS group compared to either the LPS or PCN group (Figure 4B). Notably, PCN stimulation triggered more increased expression of TLR4, MyD88, and TNF-

α than those in LPS-treated cells. Moreover, we examined the immunofluorescence staining of phosphor-p65 in cells with different treatments, and the data are shown in Figure 4C. There was a strong staining of p65 in both the PCN and LPS groups as compared to the control group. Moreover, the combination of PCN and LPS strengthened the staining of phosphor-p65 in cells compared to those in the PCN or LPS group.

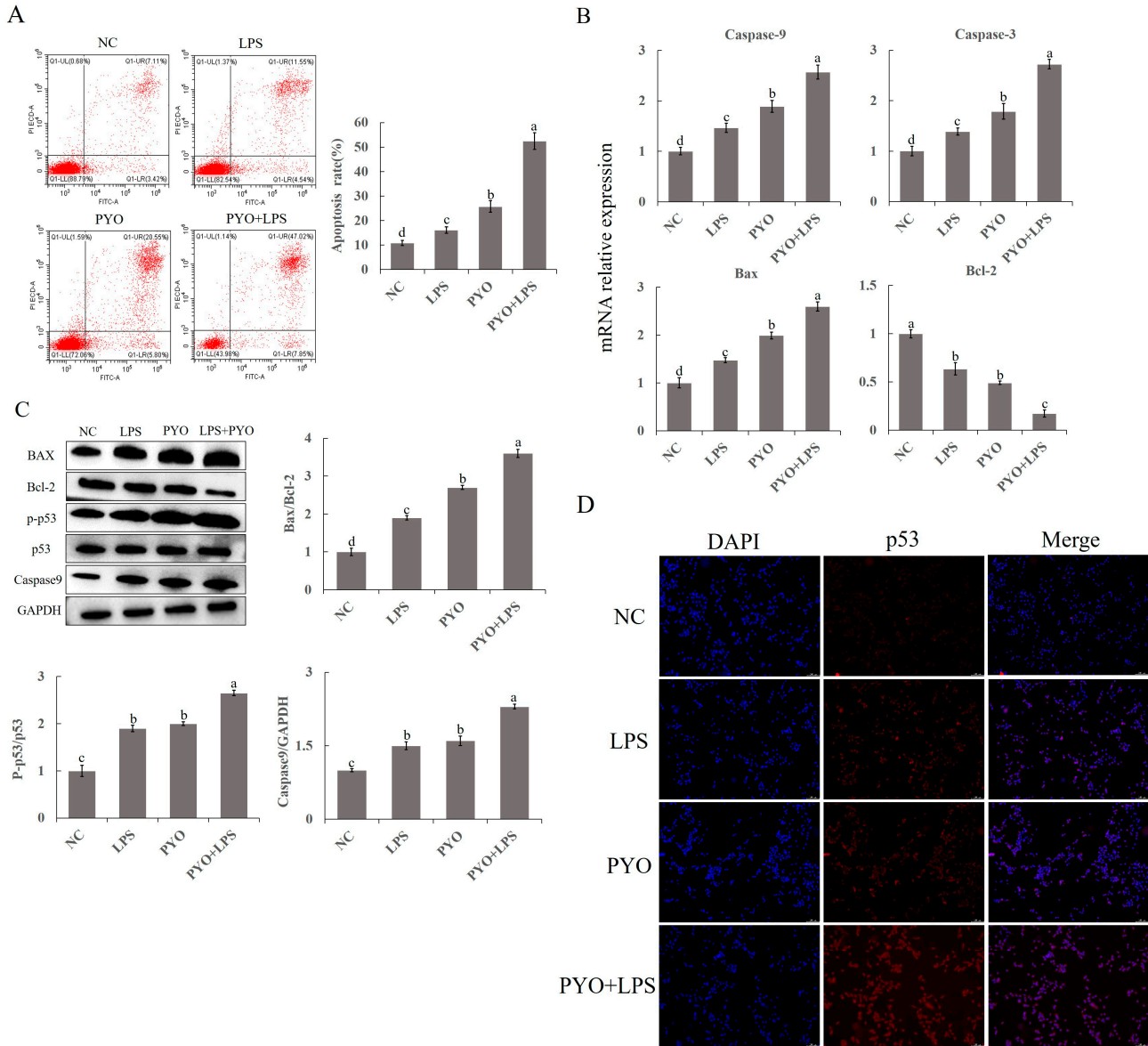

**Figure 3.** The apoptotic effect of PCN combined with LPS on bMECs. (**A**) Optotic cells were examined using flow cytometry. The apoptotic rate was calculated with the value of early—and late—apoptotic cells. (**B**) The mRNA expression of p53 signaling genes. The abundance of each gene was normalized by the geometric mean of the internal control genes (GAPDH, UXT, and RPS9). The abundance of genes in the NC group was set at 1.0. (**C**) Immunoblots were determined for the expression of proteins related to p53 signaling. Experiments were performed as three independent replicates. (**D**) Immunofluorescence of p-p53 proteins expressed in bMECs. Imaged using DMi8 Microsystems GmbH. Cells were observed at ×100 magnification. Results are expressed as means ± SEM. Letters in superscript indicate that the difference between groups was significant ($p < 0.05$). Groups with different letters indicate significant differences between the groups ($p < 0.05$), and groups with the same letter indicate no significant differences between the groups ($p > 0.05$). PCN, pyocyanin; bMECs, bovine mammary epithelial cells; NC, negative control.

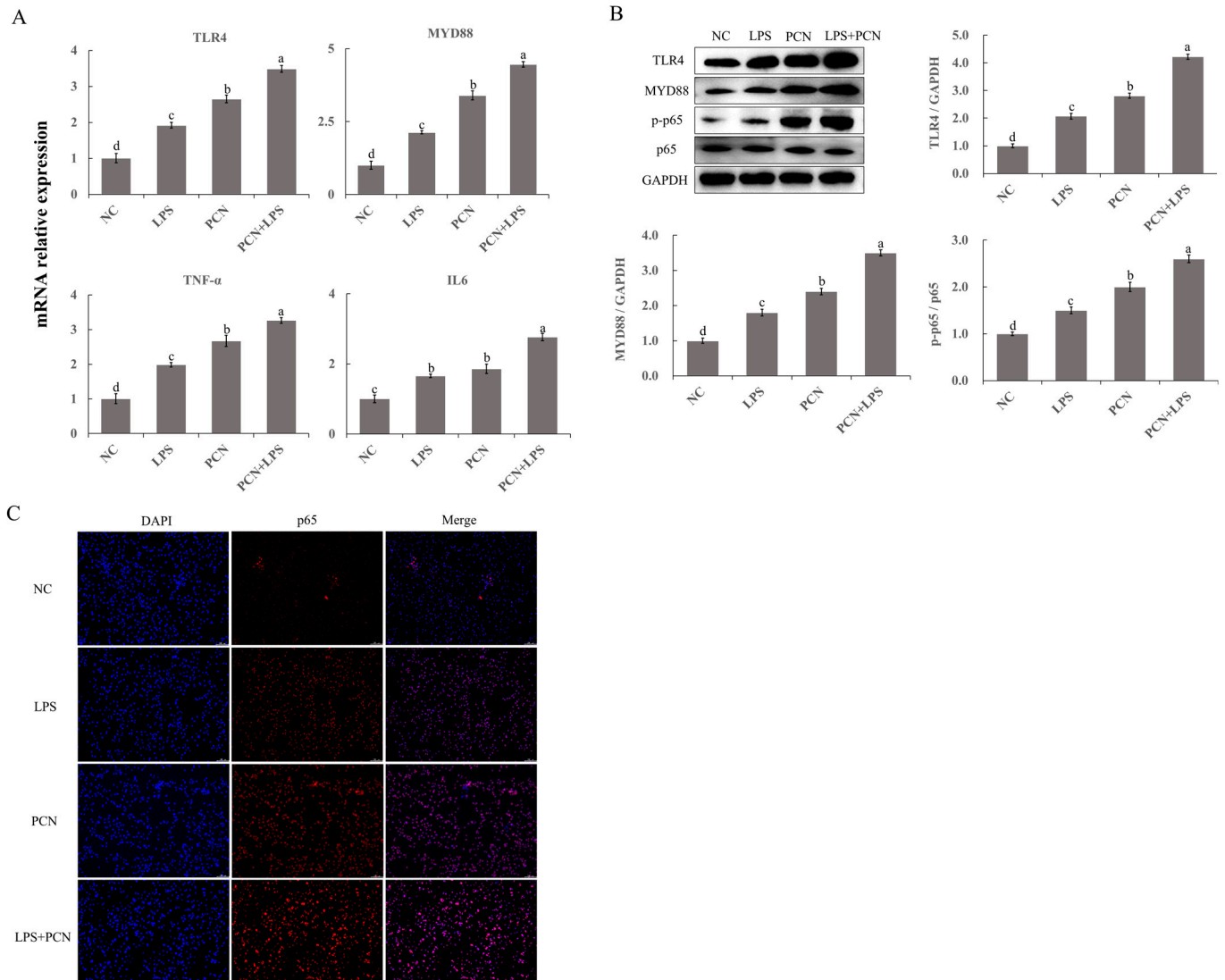

**Figure 4.** The effect of PCN treatment combined with LPS on the cellular inflammatory response. (**A**) The mRNA expression of TLR4 and NF-κB signaling genes. The abundance of each gene was normalized by the geometric mean of the internal control genes (GAPDH, UXT, and RPS9). The abundance of genes in the NC group was set at 1.0. (**B**) Immunoblots were determined for the expression of proteins related to TLR4 and NF-κB signaling. Experiments were performed as three independent replicates. (**C**) Immunofluorescence of p-p65 proteins expressed in bMECs. Imaged using DMi8 Microsystems GmbH. Cells were observed at ×100 magnification. Results are expressed as means ± SEM. Letters in superscript indicate that the difference between groups was significant ($p < 0.05$). Groups with different letters indicate significant differences between the groups ($p < 0.05$), and groups with the same letter indicate no significant differences between the groups ($p > 0.05$). PCN, pyocyanin; bMECs, bovine mammary epithelial cells; NC, negative control.

### 3.5. PCN Challenge Exacerbates the LPS-Induced Oxidative Stress in bMECs

Increased intracellular ROS staining in the LPS and PCN groups confirmed that PCN and LPS induced ROS production in bMECs (Figure 5A). However, ROS production was significantly increased in cells co-treated with LPS and PCN as compared to either the PCN or LPS group. Similarly, the addition of PCN significantly increased the accumulation of MDA compared to the NC and LPS groups ($p < 0.05$). In terms of antioxidant capacity (Figure 5B), LPS and PCN significantly inhibited GSH-Px and T-AOC activities, and the addition of PCN significantly decreased GSH-Px and T-AOC activities compared to the

control and LPS groups. The expression of the antioxidant-related genes was also detected by qPCR (Figure 5C). LPS significantly decreased the mRNA expression of HO-1 and Nrf2 compared to the control group ($p < 0.05$). However, the addition of PCN significantly decreased the expression of HO-1 and Nrf2 compared to those in the LPS group ($p < 0.05$). Consistently, proteins related to anti-oxidative stress were also altered by PCN, LPS, or a combination of PCN and LPS treatment as a result of downregulation of Nrf2 and HO-1 in the PCN, LPS, and PCN + LPS groups compared to the control group. The expression of Keap1 in the PCN, LPS, and PCN + LPS groups was significantly up-regulated compared with the control group since Keap1 is an inhibitor of Nrf2 activation (Figure 5C,D). Moreover, cells in the PCN + LPS group had the lowest antioxidant capacity among the groups (Figure 5D).

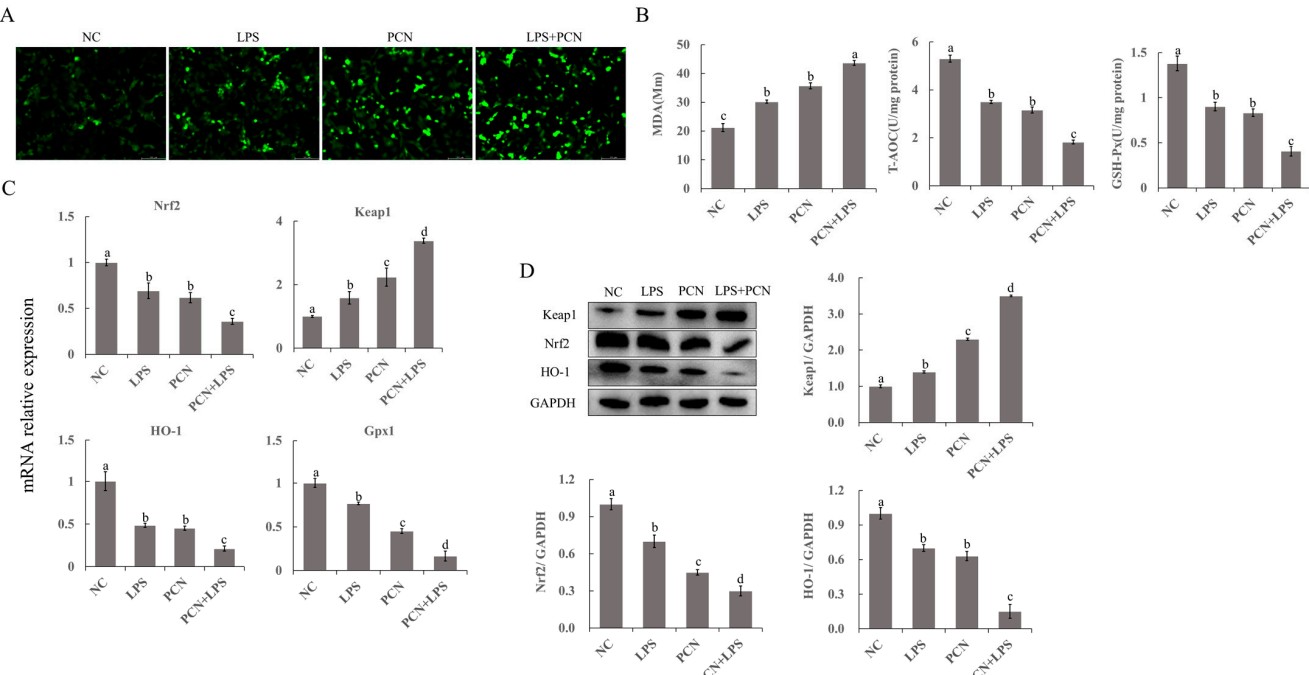

**Figure 5.** The effect of co-treatment with PCN and LPS on the cellular oxidative stress response. (**A**) DCFH-DA probe for intracellular ROS content was captured and determined. Cells were observed at ×200 magnification. (**B**) The absorbance of intensity for MDA, GSH-Px, and T-AOC was obtained at 532, 420, and 520 nm, respectively. (**C**) The expression of genes related to antioxidants and Nrf2 signaling. The abundance of each gene was normalized by the geometric mean of the internal control genes (GAPDH, UXT, and RPS9). The abundance of genes in the NC group was set at 1.0. (**D**) Immunoblots were determined for the expression of proteins related to antioxidants and Nrf2 signaling. Experiments were performed independently in triplicate. Results are expressed as means ± SEM. Letters in superscript indicate that the difference between groups was significant ($p < 0.05$). Groups with different letters indicate significant differences between the groups ($p < 0.05$), and groups with the same letter indicate no significant differences between the groups ($p > 0.05$). PCN, pyocyanin; DCFH-DA, 2′,7′-dichlorofluorescein diacetate; ROS, reactive oxygen species; MDA, malondialdehyde; GSH-Px, glutathione peroxidase; T-AOC, total antioxidative capacity; NC, negative control.

## 4. Discussion

Previously, it was found that *Pseudomonas aeruginosa*, a Gram-negative bacterium that secretes a variety of virulence factors in addition to LPS, such as PCN, was isolated from mastitis milk samples [32]. In this study, PCN promoted *E. coli* adhesion on bMECs, and PCN interacted with LPS to accelerate apoptosis of bMECs and exacerbate cellular inflammation and oxidative stress.

It was found that apoptosis is generally initiated via a variety of pathways [33–35]. Mitochondria are the center of control of cellular activities, not only for the cellular res-

piratory chain and oxidative phosphorylation but also for apoptosis regulation [36]. The mitochondrial apoptotic pathway is mainly mediated by Bcl-2 family proteins. When various apoptotic stimuli are applied to mitochondria, the pro-apoptotic members of the Bcl-2 family will be activated, leading to an imbalance between pro- and anti-apoptotic proteins. The imbalance is suggested to alter the permeability of the mitochondrial membrane, result in a loss of transmembrane potential, and finally result in the release of cytochrome C. The release of Cyt C is a key step in the apoptotic pathway of mitochondria, and the released Cyt C will bind with Apaf-1 (Apoptotic protease activating factor-1) to form a complex and thereafter activate Caspase-9. Activated Caspase-9 can activate other Caspases, such as Caspase-3 and Caspase-7, to induce apoptosis [37,38]. The results of this study showed that the addition of PCN resulted in a significant increase in the relative mRNA and protein expression of Bax/Bcl-2 and the downstream apoptosis executors Caspase9 and Caspase3 compared to the control and LPS groups. The results suggested that the combination of PCN and LPS caused a disruption of the dynamic equilibrium between Bcl-2 and Bax, which resulted in the apoptosis of mammary epithelial cells.

It was reported that p53 in the nucleus is involved in apoptosis [39]. Moreover, p53 in the cytoplasm also promotes apoptosis through its direct association with a number of Bcl-2 family proteins in the mitochondria [40]. In the present study, immunofluorescence and immunoblotting results show that the addition of PCN significantly enhanced the expression of p53 compared with the control and LPS groups, revealing that PCN combined with LPS accelerated the apoptotic process of mammary epithelial cells through the p53 signaling pathway.

As reported, when LPS is distributed in the circulation of the host, the innate immune system responds, mainly by producing inflammatory cytokines. The pro-inflammatory cytokines play an important role in the pathogenesis of bovine mastitis and regulating the occurrence of mastitis [41]. Bovine mammary epithelial cells stimulated by LPS and PCN are associated with a number of inflammatory cytokines, such as TNF-$\alpha$, IL-1$\beta$, and IL-6 [42,43]. This study revealed that the addition of PCN potentiated LPS-induced inflammatory cytokine secretion in bMECs.

TLR4 is known as one of the most prominent receptors mediating the LPS response. When TLR4 binds to LPS ligands, the signal is transduced to the TIR (Toll/IL-1 receptor) region, which then further activates the NF-$\kappa$B and MAPK pathways [44,45]. When NF-$\kappa$B is activated and translocated to the nucleus by binding to the promoters of target genes, it can regulate the expression of cytokines such as IL-6, IL-18, and TNF-$\alpha$ [46]. When cells are stimulated by LPS, the I$\kappa$B kinase complex is activated, and NF-$\kappa$B is released by phosphorylated I$\kappa$B. Free NF-$\kappa$B enters the nucleus and initiates the transcription of immune genes [47]. PCN was shown to regulate the expression of inflammatory factors through protein kinase C (PKC), mitogen-activated protein kinases (MAPKs), and NF-$\kappa$B signaling pathways [25]. In this study, we found that the addition of PCN significantly increased the expression of TLR4, p65, and p-p65 compared to the control and LPS groups, demonstrating that PCN induces cellular inflammation through the TLR4/NF-$\kappa$B pathway and enhances the degree of inflammation by interacting with LPS.

Oxidative stress leads to cell death and irreparable cell damage, and excessive production of ROS leads to cellular oxidative stress, which in turn causes oxidative stress damage to cells [48]. Some studies have reported that bovine mastitis is associated with oxidative stress. Oxidative damage to the mammary gland is associated with increased serum levels of oxidative stress-related markers such as MDA and nitric oxide (NO) [49]. In this study, the addition of PCN significantly increased cellular ROS levels and MDA levels and lowered GSH-Px and T-AOC activities compared to the control and LPS groups. The results suggested that PCN is more likely to induce cellular oxidative stress than LPS is, and that cellular oxidative stress is more severe when PCN and LPS are combined.

Nrf2 is a transcriptional activator involved in the regulation of cellular redox homeostasis and is important in the regulation of cellular oxidative stress responses [50]. HO-1 is a gene downstream of Nrf2, and when Nrf2 is activated by oxidative stress, it enters the

nucleus and initiates the transcription and expression of HO-1, which catalyzes the degradation of hemoglobin and carbon monoxide, and inhibits oxidative stress damage [51]. The results of the present study indicated that both PCN and LPS significantly promoted the oxidative stress of cells by inhibiting the Nrf2 pathway. ROS are by-products of mitochondrial physiological metabolism and can be involved in a variety of cell signaling pathways as well as tissue injury and pathophysiological processes [52]. It was reported that ROS can induce apoptosis through the death receptor pathway [53]. The present study did not consider the role of ROS in the apoptotic pathway, and whether PCN induces apoptosis through death receptors needs to be further analyzed.

In this study, PCN and LPS simultaneously activated NF-κB and p53 signaling pathways and inhibited Nrf2 signaling pathways to induce inflammation, apoptosis, and oxidative stress responses. However, there were differences in their induction of inflammation, apoptosis, and oxidative stress responses due to the different degrees of activation or inhibition of signaling pathways. However, it cannot be ruled out that PCN alone or combined with LPS can induce inflammation, apoptosis, and oxidative stress through additional pathways. For example, studies have shown that PCN can bind to aromatic receptors (AhR) to induce neutrophilic inflammation [54]. Therefore, PCN may additionally activate the AhR signal transduction pathway and induce inflammation during the stimulation of mammary epithelial cells, but this needs further proof.

## 5. Conclusions

In conclusion, PCN manifested its role in promoting apoptosis, inflammation, and oxidative stress and in collaboration with LPS to enhance the more serious biological stress responses. Through this experiment, we guessed that *Pseudomonas aeruginosa* may damage the mammary glands of dairy cows more than other Gram-negative bacteria, but this needs to be further proven.

**Supplementary Materials:** The following supporting information can be downloaded at: https://www.mdpi.com/article/10.3390/agriculture13122192/s1, Table S1: Primers used for real-time qPCR.

**Author Contributions:** Methodology, H.Z. and Y.H.; Validation, W.C.; Formal analysis, W.C.; Investigation, H.Z., W.C. and Y.H.; Data curation, H.Z.; Writing—original draft, H.Z.; Writing—review & editing, N.A.K. and Z.Y.; Supervision, N.A.K.; Project administration, Z.Y.; Funding acquisition, Z.Y. All authors have read and agreed to the published version of the manuscript.

**Funding:** This study was supported by the National Natural Science Foundation of China (32372847).

**Institutional Review Board Statement:** The manuscript does not contain experiments using animals or human studies.

**Data Availability Statement:** The data presented in this study are available on request from the corresponding author.

**Acknowledgments:** The authors would like to thank all those who helped us during this project for their valuable advice and support.

**Conflicts of Interest:** The authors declare that the research was conducted in the absence of any commercial or financial relationships that could be construed as a potential conflict of interest.

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
