# Peer review of "Involvement of Pyocyanin in Promoting LPS-Induced Apoptosis, Inflammation, and Oxidative Stress in Bovine Mammary Epithelium Cells"

_agriculture, doi:10.3390/agriculture13122192_

Round 1

Reviewer 1 Report

Comments and Suggestions for Authors

Review report: agriculture-2706288

The manuscript submitted to Agriculture, entitled “Involvement of pyocyanin in promoting LPS induced apoptosis, inflammation, and oxidative stress in bovine mammary epithelium cells” discuss the immune inflammatory and cell death induced by the addition of LPS  in conjunction with  pyocyanin (PYO)  in bovine epithelium cell lines.  The study showed the PYO exacerbates the effects normally associated with LPS and this may infer that co-infection of E. coli and Pseudomonas may be the cause of necrosis of cells observed for mastitis.  Understanding these mechanisms may assist in employing preventative treatments in future. 

This paper will be of general interest to researchers in the field, but does require extensive language editing and some revisions.

Listed below are some comments and suggestions for its improvement:

1.       Introduce the abbreviations used and use the abbreviation there after. PYO and LPS in line 14 is not introduced in the abstract and the same abbreviation is used for the experimental group name which are confusing.

a.       Line 69 start with PYO….. but line 84 “Pyocyanin” is used and should be for line 69: Pyocyanin (PYO) and then line 84 should start with PYO

b.       The abbreviation bMECs in line 79 is also not introduced

c.       Please also correct the rest in the paper

2.       Line 17 change “dose-dependent experiments, In this study” to dose-dependent experiments. In this study…”

3.       Line 43: use host resistance instead of “cow” resistance

4.       Line 69-71: rephrase “ are easily crossed in cells” it is not clear what this is referring too?

5.       Line 73: please indicate the origin of these epithelial cells? Are they also bovine cells or mouse or human?

6.       Line 99: change to “complete medium” instead of “whole medium”

7.       Line 105: please give more detail on the pHrodo green probe used – what does this bind to?

8.       Line 110: rephrase – it is unclear with is meant with “start tapping”

9.       Methods in general add the manufacturer to all products used for example line 117 “CCK-8 kit”

10.   Line 125: Indicate if the Annexin V-FITC/PI Apoptosis Detection Kit is cross-reactive with bovine cells also all the antibodies listed in lines 152-155

11.   Line 160: Please clarify which cells were used for these assays

12.   Line 178-179: Rephrase? Increased green fluorescently labelled E. coli attached to bMEC were observed, indicating that PYO increase attachment? Is this dose dependent? Were higher doses tested?

13.   Figure 1B and all other figures: please clarify “The letters in superscript indicate that the difference between groups was significant 191 (p < 0.05).”  Which groups were compared and why were different letters used in the same figure and sometimes they are repeated? In figure 2 this explanation was not added to the legend.

14.   Line 319-323: explains that an increase in the Bcl-2 was observed and it is stated that this induce Caspase 9 and 3 and apoptosis? But in the next paragraph it is stated that Bcl-2 is “anti-apoptotic” ? Please rephrase these to show that Bcl-2 is a regulator of apoptosis i.e. is in fact anti-apoptotic but elevated levels of this protein does not induce apoptosis but is produced to regulate the effects of caspase 3 (to inhibit apoptosis of healthy cells). It is known that in some cases Bcl2 and caspase 3 over expression is rather an indication of pathogenesis in some diseases.

Comments on the Quality of English Language

There is numerous translational errors for example : used "whole" medium instead of complete medium etc. some is outlined in the report.

Author Response

The manuscript submitted to Agriculture, entitled “Involvement of pyocyanin in promoting LPS induced apoptosis, inflammation, and oxidative stress in bovine mammary epithelium cells” discuss the immune inflammatory and cell death induced by the addition of LPS  in conjunction with  pyocyanin (PYO)  in bovine epithelium cell lines.  The study showed the PYO exacerbates the effects normally associated with LPS and this may infer that co-infection of E. coli and Pseudomonas may be the cause of necrosis of cells observed for mastitis.  Understanding these mechanisms may assist in employing preventative treatments in future. This paper will be of general interest to researchers in the field, but does require extensive language editing and some revisions.

Comment 1: Introduce the abbreviations used and use the abbreviation there after. PYO and LPS in line 14 is not introduced in the abstract and the same abbreviation is used for the experimental group name which are confusing.

  1. Line 69 start with PYO….. but line 84 “Pyocyanin” is used and should be for line 69: Pyocyanin (PYO) and then line 84 should start with PYO
  2. The abbreviation bMECs in line 79 is also not introduced
  3. Please also correct the rest in the paper

RE: Thanks for the comments. We have corrected them in the revised manuscript.

Comment 2: Line 17 change “dose-dependent experiments, In this study” to dose-dependent experimentsIn this study…”

RE: Thanks. We have revised it in the revised manuscript.

Comment 3: Line 43: use host resistance instead of “cow” resistance

RE: Thanks. We have rewritten the passage in the revised manuscript.

Comment 4: Line 69-71: rephrase “ are easily crossed in cells” it is not clear what this is referring too?

RE: Thanks. We have revised this sentence in the revised manuscript.

Comment 5: Line 73: please indicate the origin of these epithelial cells? Are they also bovine cells or mouse or human?

RE: Thanks. The origin of these epithelial cells are human, We have labeled it in the revised manuscript.

Comment 6:Line 99: change to “complete medium” instead of “whole medium”

RE:Thanks. We have revised it in the revised manuscript.

Comment 7: Line 105: please give more detail on the pHrodo green probe used – what does this bind to?

RE: We have given more detail on the pHrodo green probe used in the revised manuscript.

Comment 8: Line 110: rephrase – it is unclear with is meant with “start tapping”

RE: We have rephrase the sentence in the revised manuscript.

Comment 9: Methods in general add the manufacturer to all products used for example line 117 “CCK-8 kit”

RE: Thanks. We have added the manufacturer to all products in the revised manuscript.

Comment 10: Line 125: Indicate if the Annexin V-FITC/PI Apoptosis Detection Kit is cross-reactive with bovine cells also all the antibodies listed in lines 152-155.

RE: We believe that apoptosis and immunoblotting are two independent experiments without cross-reaction, and Annexin V-FITC and PI are dyes, PI can penetrate the cell membrane and stain the nucleus red, Annexin V-FITC is bound to the phosphatidylserine on the inside of the cell membrane

Comment 11: Line 160: Please clarify which cells were used for these assays

RE: We have clarified bMECs were used for these assays in the revised manuscript.

Comment 12: Line 178-179: Rephrase? Increased green fluorescently labelled E. coli attached to bMEC were observed, indicating that PYO increase attachment? Is this dose dependent? Were higher doses tested?

RE: We have redescribed the results of the experiment. After the addition of 1 and 2 μg/mL PCN, the number of E. coli attached to bMECs increased significantly, indicating that PCN can promote the adhesion of E. coli, but whether it is dose-dependent cannot be explained by the results we provided.

Comment 13: Figure 1B and all other figures: please clarify “The letters in superscript indicate that the difference between groups was significant 191 (p < 0.05).”  Which groups were compared and why were different letters used in the same figure and sometimes they are repeated? In figure 2 this explanation was not added to the legend.

RE: First, the mean values of each group in each experiment are ranked from largest to smallest, and then the largest mean is labeled with the letter a; And compare the mean with the following means, where the difference is not significant, are marked with the letter a, until a significant difference in the mean, marked with the letter b; This average, marked b, is then taken as the standard, and compared with the above averages, which are larger than it, and all that are not significant are also marked with the letter b; The maximum average marked b is then taken as the standard, and the following unmarked averages are compared, and all non-significant ones continue to be marked with the letter b until they encounter a significantly different average marked c.

The presence of repeated letters indicates that there is no significant difference between groups.

We have added the explanation to the legend in Figure 2

Comment 14: Line 319-323: explains that an increase in the Bcl-2 was observed and it is stated that this induce Caspase 9 and 3 and apoptosis? But in the next paragraph it is stated that Bcl-2 is “anti-apoptotic” ? Please rephrase these to show that Bcl-2 is a regulator of apoptosis i.e. is in fact anti-apoptotic but elevated levels of this protein does not induce apoptosis but is produced to regulate the effects of caspase 3 (to inhibit apoptosis of healthy cells). It is known that in some cases Bcl2 and caspase 3 over expression is rather an indication of pathogenesis in some diseases.

RE: Thanks for the comment. In the article, we pointed out that the gene expression and protein expression of Bax/Bcl-2 were significant. Bax is a pro-apoptotic gene, Bcl-2 is an anti-apoptotic gene, and we described them in combination. In Line 319-323, we do not describe the increase in Bcl-2.

Reviewer 2 Report

Comments and Suggestions for Authors

The study investigates at the role of pyocyanin in promoting LPS-induced apoptosis, inflammation, and oxidative stress in bovine mammary epithelial cells. The work is well-written in general. However, some of the following points must be assessed and corrected. Please take into consideration all of the suggestions provided below. Please elaborate on the association between PYO and LPS, as well as cellular immune responses and the mechanism, in the introduction. What underlying processes are responsible for the alterations that promote apoptosis, inflammation, and oxidative stress, as well as interacting with LPS to exacerbate more significant biological stress responses?  The fundamental causes of these changes must be discussed in the discussion section.   

Author Response

Comment: The study investigates at the role of pyocyanin in promoting LPS-induced apoptosis, inflammation, and oxidative stress in bovine mammary epithelial cells. The work is well-written in general. However, some of the following points must be assessed and corrected. Please take into consideration all of the suggestions provided below. Please elaborate on the association between PYO and LPS, as well as cellular immune responses and the mechanism, in the introduction. What underlying processes are responsible for the alterations that promote apoptosis, inflammation, and oxidative stress, as well as interacting with LPS to exacerbate more significant biological stress responses?  The fundamental causes of these changes must be discussed in the discussion section.   

RE: Thanks for the comments on this study. We have added the association between PCN and LPS, as well as cellular immune responses and the mechanism, in the introduction.We also analyzed which underlying processes are responsible for the alterations that promote apoptosis, inflammation, and oxidative stress, as well as interacting with LPS to exacerbate more significant biological stress responses.

Reviewer 3 Report

Comments and Suggestions for Authors

Abstract

Line 24-25: In relation to the previous context, the sentence seems confusing “In contrast, the expression of these genes and proteins was significantly upreg-24 ulated after PYO treatment combined with LPS as compared to either LPS or PYO challenge alone 25 (p<0.05)”. Please rephrase.

Introduction:

Lines 39-44: the paragraph is confusing. Please rephrase.

Methods

The experimental design in general is not well described and needs improvement

There is no description of the live bacteria used in the experiments

Line 119: Cell viability assay, what do you mean by that “cells with different concentrations of chlorpyrifos-treated 119 cells per well were incubated with 10 μl of CCK-8” ?

Line 124: what do you mean by cell differentiation ? please describe the procedure in more details.

The ROS, MDA, GSH-Px, and T-AOC detection assay is not well described. Please revise.

The lines 137-139 are confusing. Please revise “qRT-PCR was performed using HiScript Q RT SuperMix for qPCR (+gDNA wiper) (Vazyme, Nanjing, China). qRT-PCR was performed using HiScript II One Step qRT-PCR SYBR Green Kit (Vazyme, Nanjing, China) and Applied Biosystems 7300 Real-Time PCR System”.

Line 144: please correct the sentence

Line 155: were the antibodies purchased as 1:1000 dilution?

Results

Figure 1: is the title correct? What do you mean by inhibition of adherence?

Figure 2 is not clear. I don’t see any combined stimulation in the figure. Please revise.

The analysis of apoptosis must be merged with the cell vitality assay because PI-positive cells, here considered as late apoptotic, are actually dead cells.

Discussion:

Ref 28 does not support the idea cited for

1 µg/ml of LPS and PYO is high concentration and it is necrotic for the cells. Why you did not try lower concentrations.

Comments on the Quality of English Language

Extensive editing of English language is required

Author Response

Comment 1:Abstract.Line 24-25: In relation to the previous context, the sentence seems confusing “In contrast, the expression of these genes and proteins was significantly upreg-24 ulated after PYO treatment combined with LPS as compared to either LPS or PYO challenge alone(p<0.05)”. Please rephrase.

RE: Thanks for the comments. We have rephrased the sentence in the revised manuscript.

Comment 2:Introduction.Lines 39-44: the paragraph is confusing. Please rephrase

RE: Thanks. We have rephrased the paragraph in the revised manuscript.

Comment 3:Methods.The experimental design in general is not well described and needs improvement

RE: Thanks. We have redescribed the experimental design in the revised manuscript.

Comment 4: There is no description of the live bacteria used in the experiments

RE: Thanks. We added descriptions of the live bacteria used in the experiment in the revised manuscript.

Comment 5: Line 119: Cell viability assay, what do you mean by that “cells with different concentrations of chlorpyrifos-treated 119 cells per well were incubated with 10 μl of CCK-8” ?

RE: Thanks. We have revised this sentence in the revised manuscript.

Comment 6: Line 124: what do you mean by cell differentiation ? please describe the procedure in more details.

RE: Thanks. This is our description error, we have revised it in the revised manuscript.

Comment 7: The ROS, MDA, GSH-Px, and T-AOC detection assay is not well described. Please revise

RE: Thanks. We have revised it in the revised manuscript.

Comment 8: The lines 137-139 are confusing. Please revise “qRT-PCR was performed using HiScript Q RT SuperMix for qPCR (+gDNA wiper) (Vazyme, Nanjing, China). qRT-PCR was performed using HiScript II One Step qRT-PCR SYBR Green Kit (Vazyme, Nanjing, China) and Applied Biosystems 7300 Real-Time PCR System”

RE: Thanks. We have revised the paragraph in the revised manuscript.

Comment 9: Line 144: please correct the sentence

RE: Thanks. We have corrected the sentence in the revised manuscript.

Comment 10: Line 155: were the antibodies purchased as 1:1000 dilution?

RE: Thanks. We buy the antibody stock and dilute it at a ratio of 1:000. We have corrected the sentence in the revised manuscript.

Comment 11: Results. Figure 1: is the title correct? What do you mean by inhibition of adherence?

RE: Thanks. We have corrected the title in the revised manuscript .

Comment 12: Figure 2 is not clear. I don’t see any combined stimulation in the figure. Please revise.

RE: Thanks. We have revised the Figure 2 in the revised manuscript.

Comment 13:The analysis of apoptosis must be merged with the cell vitality assay because PI-positive cells, here considered as late apoptotic, are actually dead cells.

RE: Thanks. We used Annexin V and PI to distinguish between living cells, early apoptotic cells and dead cells, and the results showed that the changes of early apoptotic cells and dead cells were the same under different treatments, so the cell viability test was not incorporated

Comment 14: Ref 28 does not support the idea cited for

RE: Thanks. We have sought the new literature to support the idea in the revised manuscript.

Comment 15: 1 µg/ml of LPS and PYO is high concentration and it is necrotic for the cells. Why you did not try lower concentrations.

RE: Thanks. Since the LPS we used in our experiments were purchased from companies, and the LPS produced by each company were different, we learned from a literature that they used 1μg/mL LPS to induce inflammation, and we purchased LPS from the same company, and we found that 1μg/mL LPS was able to induce inflammation during our experiments. Because we compared the difference in inflammation induced by LPS and PCN in the experiment, 1μg/mL LPS and PCN were used to maintain the same concentration.The reference is as follows:

Fan Y, Shen J, Liu X, et al. β-Sitosterol Suppresses Lipopolysaccharide-Induced Inflammation and Lipogenesis Disorder in Bovine Mammary Epithelial Cells. Int J Mol Sci. 2023;24(19):14644

Comment 16: Extensive editing of English language is required

RE: Thanks. We have re-edited the English language

Reviewer 4 Report

Comments and Suggestions for Authors

The aim of this experiment was to compare the degree of apoptosis, inflammation and oxidative stress of dairy mammary epithelial cells induced by LPS and PYO, and to examine whether PYO can promote the apoptosis, inflammation and oxidative stress of bovine mammary epithelial cells induced by LPS.

Pseudomonas aeruginosa is an opportunistic pathogen in humans and animals. In ruminants, it is responsible for mastitis that may occur sporadically or as an outbreak within dairy herds. It can be refractory to treatment, and therefore, economic implications are high. To avoid spread within herds, it is important to identify the sources of infection or transmission ways and to implement effective prophylactic measures. Intra-mammary infections with P. aeruginosa can occur from both, a point source, or from a continuous exposure to a common contamination, e.g., from dirty, or soiled environment. Clinical signs seem to depend on the exposure dose and range from toxic mastitis with severe symptoms of toxemia, marked swelling of the mammary gland, high body temperature, and watery milk secretions with clots or blood, to chronic non-clinical infections with elevations of somatic cell counts. Due to the importance of this causative agent in the pathology of mastitis in cows, I believe that the authors have well chosen PYO as a substance whose pathophysiological action should be determined, in addition to LPS, which has a well-known role in the pathogenesis of mastitis in cows.

Pyocyanin is one of the many toxic compounds produced and secreted by the Gram negative bacterium Pseudomonas aeruginosa. Pyocyanin is a blue secondary metabolite, turning red below pH 4.9, with the ability to oxidise and reduce other molecules and therefore kill microbes competing against P. aeruginosa as well as mammalian cells. Since pyocyanin is a zwitterion at blood pH, it is easily able to cross the cell membrane. There are three different states in which pyocyanin can exist: oxidized (blue), monovalently reduced (colourless) or divalently reduced (red). Mitochondria play an important role in the cycling of pyocyanin between its redox states. Due to its redox-active properties, pyocyanin generates reactive oxygen species. Pyocyanin inactivates catalase by reducing its gene’s transcription as well as directly targeting the enzyme itself. Glutathione is an important antioxidant modulated by pyocyanin. The cell cycle can be disturbed by the action of pyocyanin, and it can hinder the proliferation of lymphocytes. This is done by the generation of reactive oxygen intermediates, such as hydrogen peroxide and superoxide, which cause oxidative stress by directly damaging DNA or by targeting other constituents of the cell cycle such as DNA recombination and repair machinery. Pyocyanin contributes to the disproportion of protease and antiprotease activity by disabling α1- protease inhibitor. The mentioned features of PYO represent the basis for the conclusions made by the authors and make them well-founded.

The standard abbreviation for pyocyanin is PCN, so maybe that abbreviation could be adopted through the text, but it is not of crucial importance for the quality of the work.

The introduction is well and concretely written. The material and methods are well chosen and applied. The results are well presented. The discussion is well written. The selection of references is adequate. The conclusion is in the aims of the manuscript.

Author Response

Comment :The aim of this experiment was to compare the degree of apoptosis, inflammation and oxidative stress of dairy mammary epithelial cells induced by LPS and PYO, and to examine whether PYO can promote the apoptosis, inflammation and oxidative stress of bovine mammary epithelial cells induced by LPS.

Pseudomonas aeruginosa is an opportunistic pathogen in humans and animals. In ruminants, it is responsible for mastitis that may occur sporadically or as an outbreak within dairy herds. It can be refractory to treatment, and therefore, economic implications are high. To avoid spread within herds, it is important to identify the sources of infection or transmission ways and to implement effective prophylactic measures. Intra-mammary infections with P. aeruginosa can occur from both, a point source, or from a continuous exposure to a common contamination, e.g., from dirty, or soiled environment. Clinical signs seem to depend on the exposure dose and range from toxic mastitis with severe symptoms of toxemia, marked swelling of the mammary gland, high body temperature, and watery milk secretions with clots or blood, to chronic non-clinical infections with elevations of somatic cell counts. Due to the importance of this causative agent in the pathology of mastitis in cows, I believe that the authors have well chosen PYO as a substance whose pathophysiological action should be determined, in addition to LPS, which has a well-known role in the pathogenesis of mastitis in cows.

Pyocyanin is one of the many toxic compounds produced and secreted by the Gram negative bacterium Pseudomonas aeruginosa. Pyocyanin is a blue secondary metabolite, turning red below pH 4.9, with the ability to oxidise and reduce other molecules and therefore kill microbes competing against P. aeruginosa as well as mammalian cells. Since pyocyanin is a zwitterion at blood pH, it is easily able to cross the cell membrane. There are three different states in which pyocyanin can exist: oxidized (blue), monovalently reduced (colourless) or divalently reduced (red). Mitochondria play an important role in the cycling of pyocyanin between its redox states. Due to its redox-active properties, pyocyanin generates reactive oxygen species. Pyocyanin inactivates catalase by reducing its gene’s transcription as well as directly targeting the enzyme itself. Glutathione is an important antioxidant modulated by pyocyanin. The cell cycle can be disturbed by the action of pyocyanin, and it can hinder the proliferation of lymphocytes. This is done by the generation of reactive oxygen intermediates, such as hydrogen peroxide and superoxide, which cause oxidative stress by directly damaging DNA or by targeting other constituents of the cell cycle such as DNA recombination and repair machinery. Pyocyanin contributes to the disproportion of protease and antiprotease activity by disabling α1- protease inhibitor. The mentioned features of PYO represent the basis for the conclusions made by the authors and make them well-founded.

The standard abbreviation for pyocyanin is PCN− , so maybe that abbreviation could be adopted through the text, but it is not of crucial importance for the quality of the work.

The introduction is well and concretely written. The material and methods are well chosen and applied. The results are well presented. The discussion is well written. The selection of references is adequate. The conclusion is in the aims of the manuscript.

RE:  Thanks so much for your comments and suggestions. Also, we have changed the abbreviation of pyocyanin in the whole article.

Round 2

Reviewer 1 Report

Comments and Suggestions for Authors

Check spacing before “(“

The following concerns from the previous review were not addressed:

 1.       Line 125: Indicate if the Annexin V-FITC/PI Apoptosis Detection Kit are cross-reactive with bovine cells also all the antibodies listed in lines 152-155.  Indicate if these antibodies will bind to bovine protein? Are they specific for bovine? Give appropriate references

2.       The antibodies listed in Lines 154-156 are not specific for bovine but are used for mouse or human. Please indicate if this have been tested to also cross-react (bind) to bovine proteins?

3.       Figure 1B and all other figures: please clarify “The letters in superscript indicate that the difference between groups was significant 191 (p < 0.05).”  Which groups were compared and why were different letters used in the same figure and sometimes they are repeated? 

Comments on the Quality of English Language

Improved 

Author Response

Comment 1. Check spacing before “(“

RE: Thanks. We have revised it in the revised manuscript.

Comment 2. Line 125: Indicate if the Annexin V-FITC/PI Apoptosis Detection Kit are cross-reactive with bovine cells also all the antibodies listed in lines 152-155.  Indicate if these antibodies will bind to bovine protein? Are they specific for bovine? Give appropriate references

RE: In early apoptotic cells, annexin V binds to phosphatidylserine (PS) exposed outside the bovine cells. The Annexin V-FITC/PI Apoptosis Detection Kit are not cross-reactive with all the antibodies listed in lines 152-155. We have validated these antibodies in previous experiments, and they bind specifically to bovine proteins.

Reference 1: Xu T, Zhu H, Liu R, et al. The protective role of caffeic acid on bovine mammary epithelial cells and the inhibition of growth and biofilm formation of Gram-negative bacteria isolated from clinical mastitis milk. Front Immunol. 2022; 13:1005430. 

Reference 2: Xu T, Liu R, Zhu H, Zhou Y, Pei T, Yang Z. The Inhibition of LPS-Induced Oxidative Stress and Inflammatory Responses Is Associated with the Protective Effect of (-)-Epigallocatechin-3-Gallate on Bovine Hepatocytes and Murine Liver. Antioxidants (Basel). 2022;11(5):914.

Comment 3. The antibodies listed in Lines 154-156 are not specific for bovine but are used for mouse or human. Please indicate if this have been tested to also cross-react (bind) to bovine proteins?

RE: We have validated these antibodies in previous experiments, and they bind specifically to bovine proteins.

Comment 4. Figure 1B and all other figures: please clarify “The letters in superscript indicate that the difference between groups was significant 191 (p < 0.05).”  Which groups were compared and why were different letters used in the same figure and sometimes they are repeated? 

RE: What this sentence (The letters in superscript indicate that the difference between groups was significant (p < 0.05))means is that groups are compared to each other, groups with different letters indicate significant differences between the groups, and groups with the same letter indicate no significant differences between the groups.

Comment 5. Comments on the Quality of English Language.  Improved 

RE: Thanks. We have Improved the quality of English Language.

Reviewer 3 Report

Comments and Suggestions for Authors

The authors addressed all the comments and improved the manuscript

Author Response

Comment 1.The authors addressed all the comments and improved the manuscript.

RE: Thank you for your comments.